# In Vitro and In Silico Activities of *E*. *radiata* and *E*. *cinerea* as an Enhancer of Antibacterial, Antioxidant, and Anti-Inflammatory Agents

**DOI:** 10.3390/molecules28207153

**Published:** 2023-10-18

**Authors:** Hayet Elkolli, Meriem Elkolli, Farid S. Ataya, Mounir M. Salem-Bekhit, Sami Al Zahrani, Mostafa W. M. Abdelmageed, Barbara Ernst, Yacine Benguerba

**Affiliations:** 1Laboratory of Multiphasic Polymeric Materials, Départment of Process Engineering, Faculty of Technology, University Ferhat Abbas of Setif 1, Setif 19000, Algeria; kolli_h@yahoo.fr; 2Laboratory of Applied Microbiology, Faculty of Natural and Life Sciences, University of Ferhat Abbas Setif 1, Setif 19000, Algeria; elkollim@yahoo.fr; 3Biochemistry Department, College of Science, King Saud University, P.O. Box 2455, Riyadh 11451, Saudi Arabia; 4Kayyali Chair for Pharmaceutical Industries, Department of Pharmaceutics, College of Pharmacy, King Saud University, P.O. Box 2457, Riyadh 11451, Saudi Arabia; 439106207@student.ksu.edu.sa; 5Department of Pharmacognosy, College of Pharmacy, King Saud University, P.O. Box 2457, Riyadh 11451, Saudi Arabia; wmostafa471@gmail.com; 6Laboratory of Molecular Recognition and Separation Processes (RePSeM), CNRS, IPHC UMR 7178, University of Strasbourg, ECPM 25 Becquerel Road, F-67000 Strasbourg, France; 7Laboratory of Biopharmacy and Pharmacotechnics (LPBT), University of Ferhat Abbas Setif 1, Setif 19000, Algeria

**Keywords:** antibacterial activity, antioxidant activity, anti-inflammatory activity, *Eucalyptus radiata*, *Eucalyptus cinerea*

## Abstract

*Eucalyptus*, a therapeutic plant mentioned in the ancient Algerian pharmacopeia, specifically two species belonging to the *Myrtaceae* family, *E*. *radiata* and *E*. *cinerea*, were investigated in this study for their antibacterial, antioxidant, and anti-inflammatory properties. The study used aqueous extracts (AE) obtained from these plants, and the extraction yields were found to be different. The in vitro antibacterial activity was evaluated using a disc diffusion assay against three typical bacterial strains. The results showed that the two extracts were effective against all three strains. Both extracts displayed significant antioxidant activity compared to BHT. The anti-inflammatory impact was evaluated using a protein (BSA) inhibition denaturation test. The *E*. *radiata* extract was found to inhibit inflammation by 85% at a concentration of 250 µg/mL, significantly higher than the Aspirin. All phytoconstituents present good pharmacokinetic characteristics without toxicity except very slight toxicity of terpineol and cineol and a maximum binding energy of −7.53 kcal/mol for its anti-TyrRS activity in silico. The study suggests that the extracts and their primary phytochemicals could enhance the efficacy of antibiotics, antioxidants, and non-steroidal anti-inflammatory drugs (NSAIDs). As pharmaceutical engineering experts, we believe this research contributes to developing natural-based drugs with potential therapeutic benefits.

## 1. Introduction

*Eucalyptus* is a member of the *Myrtaceae* family [1,2] and is native to Australia [3,4]. It is known for its adaptability to different environmental conditions, high productivity, and effortless harvesting, making it a sustainable feedstock supply for phytoconstituents [5,6]. The plant contains a variety of volatile and non-volatile chemicals with diverse biological functions [7]. Extracts from the leaves have been reported to contain phenolics and flavonoids with antioxidant and antimicrobial properties [8], while constituents such as alkaloids, polyphenols, and propanoids exhibit anti-inflammatory, antibacterial, and antiseptic properties [9].

Although most research on the phytochemistry of *Eucalyptus* has focused on the plant’s essential oils, many non-volatile compounds, including triterpenoids, flavonoids, and tannins, have been isolated from this genus [7]. However, little information is available regarding the potential bioactivities of crude extracts from *Eucalyptus*, particularly traditional preparations [7].

To address this gap in knowledge, researchers used decoctions as it is the most usual way people take it [7]. Given current ethical concerns about animal testing, the researchers opted to use in vitro and in silico methods to investigate the antibacterial, antioxidant, and anti-inflammatory properties of aqueous extracts of two species of *Eucalyptus* [10,11]. In silico methods such as virtual screening can be used to examine plant compounds’ receptor interactions quickly and cost-effectively [11]. The study focused on two species of *Eucalyptus* cultivated in Algeria, for which no reports on the bioactivities of the aqueous extract of the leaves have been found [7].

## 2. Results

### 2.1. Extraction Yield

In this context, “yield” refers to the ratio of the amount of extract collected to the total mass of the plants used. The percentages of *E*. *radiata* and *E*. *cinerea* recovered during the extraction were 27.83 and 4.84%, respectively.

#### Antibacterial Activity

It was shown that both extracts had substantial antibacterial activity against the tested microorganisms (Table 1). Even so, *E*. *cinerea* demonstrated a zone of inhibition of 22.6 mm in diameter against *S*. *aureus*, which is a remarkable result. *P*. *aeruginosa*, on the other hand, was resistant from the 100 mg/mL concentration. Therefore, only its minimal diameters at 200 and 150 mg/mL were displayed.

### 2.2. Antioxidant Activity

Globally, the two AEs have antioxidant activity. The reduction in DPPH was dose-dependent (Figure 1) with an IC_50_ of 0.19 + 0.03 mg/mL by *E*. *radiata* and 0.15 ± 0.08 mg/mL by *E*. *cinerea* compared to the BHT (0.94 ± 0.37 mg/mL).

### 2.3. Anti-Inflammatory Test

An excellent inhibitory effect on protein denaturation was observed (Figure 2), but no dose-response relationship could be established. The extract from *E*. *radiata* was more effective than that from *E*. *cinerea* AE in preventing protein denaturation (85.21–85.91% inhibition). At a concentration of 250 µg/mL, the most common NSAID, aspirin, had an inhibition of 110.09%. Statistically, the Tukey’s multiple comparison test revealed that aspirin was significantly more active than the two extracts at a *p*-value < 0.05.

### 2.4. Docking Results

The scoring phytocompounds and gentamicin are summarized in Table 2. The ellagic acid recorded the best binding energy (−7.53 kcal/mol), but it was still less than gentamicin scoring (−10.04 kcal/mol) and that of the native ligand.

The interactive amino acids of the protein target with phytocompounds are listed as follows (Table 3):

Figure 3 presents the interactions with the molecular surface around the studied phytocompounds, as ligands, at the binding site of TyrRS. The pink area shows the electron donor region, and the green represents the electron acceptor region. The 2-D structures show the amino acids involved in the active site of the enzyme as well as the nature of the bonds established.

### 2.5. Drug Likeness and ADME Prediction

As a result of the drug-likeness filters, expensive late-stage preclinical and clinical failure can be avoided, allowing for earlier medication development [12], phytocompounds were evaluated for their drug-like characteristics using the SWISS ADME web-based tool, and it was discovered that all compounds followed Lipinski’s rule of five, much like gentamicin (Table 4). The other calculated parameters also show that all compounds were water soluble with high intestinal absorption. Ellagic acid and epicatechin had no toxic risks, while α-terpineol could be moderately irritant. However, 1,8-cineol presented mutagenic and reproductive risks.

## 3. Materials and Methods

### 3.1. The Choice of Plants

An ethnopharmacological survey (on anti-infectious plants) preceded this work spread over four months and covered all the communes of Setif (northeastern Algeria). A random sample of 75 interviewed people aged between 20 and 67. *Eucalyptus* (8%) after *Origanum* 11% and Ginger 10% among the most used plants (Figure 4). Furthermore, leaves represented 48% of plant parts used. A decoction is the most used preparation mode, with the highest rate at 60%.

In March 2017, young leaves of *E*. *radiata* (Figure 5) were harvested from cultivated plants on the Setif 1 university campus. After that, they are dried in the shade at room temperature. A neighborhood herbalist was the source from which we acquired the *E*. *cinerea* leaves (Figure 5). Dr. Nouioua provided the botanical confirmation of the species from the Department of Ecology in the FNLS of Sétif 1 University. A temperature of four degrees Celsius is maintained on the plants while they are being stored.

For this investigation, water was selected as the solvent due to its favorable characteristics, including being the least hazardous, least expensive, and most environmentally friendly solvent. The decoction as a conventional extraction method was employed, which involved boiling 20 g of dried leaves with 500 milliliters of cold, distilled water and simmering the mixture for 20 min. After filtration, the aqueous extract (AE) was obtained and air-dried to obtain the final product. The yield percentage was calculated by the ratio between the weight of the obtained dry extract and the initial weight of the plant.
(1)$$\mathrm{Yield}=\frac{\mathrm{The}\;\mathrm{weight}\;\mathrm{of}\;\mathrm{the}\;\mathrm{obtained}\;\mathrm{dried}\;\mathrm{xtract}}{\mathrm{the}\;\mathrm{weight}\;\mathrm{of}\;\mathrm{the}\;\mathrm{initial}\;\mathrm{plant}\;\mathrm{material}\;\mathrm{used}\;}\times 100$$

### 3.2. Antibacterial Activity

Three ATCC bacterial strains, including *Escherichia coli* (ATCC 25922), *Pseudomonas aeruginosa* (ATCC 27853), and *Staphylococcus aureus* (ATCC 25922), were tested for antibacterial activity using the disc diffusion technique [13]. Filter paper discs impregnated with 50, 100, 150, and 200 mg/mL of the AE are placed on Mueller-Hinton Agar (MHA) inoculated with a 0.5 Mc Farland (10^8^ cell/mL) standard inoculum and then incubated at 37 °C for 24 h. As a reference, we employed gentamicin. As a result, we can calculate the diameter of the inhibition zones for each disc.

### 3.3. Antioxidant Activity

The antioxidative activity may be tracked by measuring the rate at which the DPPH radical’s absorbance at 517 nm decreases. The method used was that of [14] with slight modifications. First, 50 µL of each AE, at varying concentrations, was mixed with 1250 µL of a methanolic solution of DPPH at 0.004%. After 30 min of incubation at room temperature in the dark, the absorbance was recorded at 517 nm. Butylated hydroxytoluene (BHT) was the standard. Thus, it was treated the same way. As a result, we may calculate the DPPH scavenging activity as follows:(2)$$\%\;\mathrm{of}\;\mathrm{DPPH}\;\mathrm{scavenging}\;\mathrm{effect}=\frac{\left(\mathrm{At}-\mathrm{Ac}\right)}{\mathrm{Ac}}$$
where At represents the absorbance of the test and Ac represents the absorbance of the reference.

### 3.4. In-Vitro Anti-Inflammatory Test

Protein denaturation inhibition is measured using a modified approach version [15]. When 100 µL of plant extract at varying concentrations (250, 500, and 1000 µg/mL) was mixed with 500 µL of Bovine Serum Albumin (BSA at 1%, the resulting mixture is called a “standard” preparation. For the first 10 min, this combination was left at room temperature and then was heated to 51 °C for 20 min. After bringing the resultant solution to room temperature, its absorbance was measured at 660 nm. As a reference point for success, we used acetylsalicylic acid. Triplicates of the experiment were performed, and the percentage of inhibition of protein denaturation was determined as follows.
(3)$$\%\;\mathrm{Inhibition}=\left(100-\frac{A1-A2}{A0}\right)\times 100$$

In this equation, A1 represents the sample absorbance, A2 denotes the product control absorbance, and A0 represents the absorbance of the positive control (aspirin solution).

GraphPad Prism 5 version 5.03 was used to create the graphs for the ethnopharmacology survey and the in vitro activities.

### 3.5. Molecular Docking

#### Phyto-Compounds

After a thorough literature analysis, the chemicals 1,8-cineol (CID 2758), ellagic acid (CID 5281855), α-terpineol (CID 17100), and epicatechin (CID 72276) were chosen as representative compounds to carry out the molecular docking investigation (Table 5). The standard antibiotic used was gentamicin (CID 3467).

In Figure 6, the 2-D structures and functional groups of the four main compounds present in the extracts are displayed, along with gentamicin and the native ligand of the enzyme. These compounds were subjected to in silico testing to assess their ability to impede the aminoacylation process facilitated by TyrRS (PDB code 1JII). To determine their binding affinities, components of the highly effective in vitro extracts were evaluated and compared to the standard antibiotic, gentamicin. The 3-D structures of the phytocompounds and gentamicin were obtained in CID format from the PubChem database.

### 3.6. Docking Analysis and Protein Preparation

AutoDock Tools 1.5.7 was utilized to determine the binding affinity of the selected phytoconstituents to the binding site of the bacterial tyrosyl-tRNA synthetase. The program implements the gradient optimization method in its local optimization process and ranks the ligands based on the empirical binding free energy (ΔG in kcal/mol) function [12]. The 3D structures of the target enzymes were obtained in PDB format from the Protein Data Bank. PDBQT format files were generated by removing water molecules, polar hydrogen, and Kollman charges were added to protein residues, and the protein’s native ligand was eliminated (2-Amino-3-(4-Hydroxyphenyl)Propionylamino)-(2,4,5,8-Tetrahydroxy-7-Oxa-2-Aza-Bi cyclo[3.2.1]Oct-3-Yl) Acetic Acid). The Lamarckian Genetic docking algorithm was employed and executed ten times. The open-source program Babel 2.4.1 was used to convert the SDF files of the substances under investigation into PDB files. The interactions were visualized in two dimensions using Discovery Studio v.16.1.0.15350 software. H-bonds between the ligands and interacting residues are depicted as lines and balls with a distance range of Å.

### 3.7. Drug Likeness, ADME/Toxicity Prediction

Lipinski’s method was employed to assess the drug-like properties of phytocompounds, which sets limits on four specific physicochemical parameters [27]. Typically, these are the characteristics of an orally active drug: the octanol-water partition coefficient (milogP) and the number of hydrogen bond donors (n-OH and n-NH) should not exceed 5, and the number of hydrogen bond acceptors (n-ONs) should be less than 10. The molecular weight (MW) should be below 500 D, and no more than one violation should occur [28]. Molinspiration Cheminformatics online tools and SwissADME online tools were used to predict physicochemical and pharmacokinetic parameters, while OSIRIS Property Explorer online tools were used to predict toxicity risks.

## 4. Discussion

According to research studies [29,30,31], decoction is Algeria’s most commonly used method for obtaining active plant chemicals. Water is preferred over organic solvents due to its safety, affordability, environmental friendliness, and accessibility for extracting phenolic compounds [32]. Traditional healers and practitioners mostly use it as a solvent [33]. However, the boiling process involved in decoction may lead to the degradation of the medicinal components of plants [34]. It has been reported that conventional extraction at 100 °C yields the highest amount of flavonoids and phenols (active compounds). However, it is impossible to obtain the best yield of all compounds simultaneously [35], as the chemical content of the extract varies depending on the preparation method [36]. A study on the aqueous leaf extracts of *E*. *camaldulensis* revealed high levels of polyphenols, saponins, and flavonoids in both qualitative and quantitative analyses [37]. The main component of *Eucalyptus* essential oils is 1,8-cineol, a monoterpene that dissolves in water at 19.85 °C [19] and is commonly found in *Eucalyptus* species [38,39,40,41,42,43].

### 4.1. Efficacy against Bacteria

According to [44], antibiotic-resistant bacteria significantly threaten public health globally. These synthetic antibiotics, especially in developing nations, are expensive and deficient for the treatment of diseases caused by pathogenic microorganisms and have incidental effects and adulterations [45]. However, medicinal plants have yielded antimicrobial compounds that offer a new tool for combatting bacterial illnesses [1]. The presence of phytochemicals in plant fractions, such as flavonoids and polyphenols, may be responsible for the disruptive bacterial membrane effect [8]. For bacterial illnesses such as sinusitis, sore throat, angina, cough, bronchitis, and urinary tract infections, [46] recommends a decoction of *Eucalyptus* leaves in water at 10–20 g/L as a daily beverage.

Similarly, the phenolic and saponin contents of the *E*. *microcorys* aqueous extract have been credited with its antibacterial capabilities [7]. *Eucalyptus* species have revealed strong antibacterial abilities against various bacteria [47], such as *S*. *aureus*, *L*. *monocytogenes*, *Bacillus*, *K*. *pneumoniae*, *E*. *faecalis*, *P*. *aeruginosa*, *S*. *enteritidis*, and *E*. *coli*. Leaf extract from *E*. *camaldulensis* showed anti-virulence and membrane disruption actions against Gram-positive *L*. *monocytogenes* [8]. However, *P*. *aeruginosa* is frequently resistant to multiple antibiotics [48] due to its outer membrane’s low permeability [49], leading to therapeutic failures [50]. It should be noted that antibacterials also target DNA topoisomerase [51] and other bacterial proteases [52].

Antioxidants in *Eucalyptus* extracts indicate the likely presence of compounds that can interact with free radicals by acting as an electron donor or hydrogen atom donor and producing a scavenging activity [53]. *Eucalyptus* leaf extracts effectively ameliorate hydrogen peroxide-induced oxidative stress by increasing cell viability, glutathione levels, and antioxidant enzyme activity and by decreasing the production of free radicals and lipid peroxidation levels [54]. The rich phenolic content of the aqueous fraction is responsible for superior antioxidant activity [8] by capturing free radicals through hydroxyl groups to reduce oxidative stress [55]. Polyphenolic substances, such as ellagic acid and epicatechin, are particularly plentiful in the *Eucalyptus* genus and responsible for the antioxidant effects of its extracts [32]. Ellagic acid, in particular, found in *E*. *cinerea*, has been demonstrated to possess potent antioxidant activity [39] that surpasses tocopherol [18]. Additionally, flavonoids increase the likelihood of antioxidant activity in *Eucalyptus* extracts [47]. Notably, conventional extraction at 90 °C for phenols is crucial for optimizing antioxidant efficiency [35].

### 4.2. Results in Reducing Inflammation

The determination of protein denaturation is a valuable tool for detecting anti-inflammatory compounds without the need for animal testing [56]. Protein denaturation is the primary cause of inflammation [15]. It is associated with the denaturation of tissue proteins (in vivo) and the onset of inflammatory and arthritic complications due to auto-antigen production [57]. Herbal extracts, which serve as safe and novel sources, can be evaluated for their anti-inflammatory efficacy [58], as an alternative to non-steroidal anti-inflammatory drugs, which have undesirable side effects such as gastrointestinal toxicity, renal injury, hepatotoxicity, hypertension, and allergic skin reactions [59,60]. Additionally, 1,8-cineol can be used for chronic inflammation management as a potent cytokine suppressant [39]. Inhalation of 1,8-cineol prior to the ovalbumin challenge was found to reduce the levels of pro-inflammatory cytokines TNF-α and IL-1β and prevent the decrease of the anti-inflammatory cytokine IL-10 in the bronchoalveolar fluid [41]. It is speculated that the activity may be due to the polyphenolic content, including flavonoids [61], such as ellagic acid and epicatechin. The extracts in the present study exhibited higher activity at lower concentrations. In contrast, the standard drug Aspirin showed the best activity, indicating that the anti-inflammatory activity of these extracts is not related to protection against protein denaturation.

### 4.3. Investigation of Docking

The use of in silico models as a preliminary screening tool for predicting a medication’s effect on cells and aiding in experimental research trial design has potential. The predictions’ results are noteworthy and helpful in designing experiments due to the likelihood of false positives in the selected chemical leads for biological activity [11]. This approach has been successful in the pharmaceutical sector [62]. The binding affinity of phytocompounds to their respective protein targets determines their scores. A higher binding affinity is achieved with lower binding energies [12]. In our study, ellagic acid (EA) exhibited the best free binding energy (−7.53 kcal/mol) and the best specific bond energies (vdW + Hbond + Desolvation) (−8.48 kcal/mol) among the docked compounds. EA has been reported to be effective against various pathogens, including bacteria, fungi, and parasites [63], and has antioxidant and anti-inflammatory properties [64]. Polyphenols such as ellagic acid and epicatechin interact with proteins primarily through hydrophobic interactions and hydrogen bonds via their aromatic and phenolic nuclei [65], consistent with our findings (Table 3). α-Terpineol also exhibited antibacterial properties [66,67], demonstrating in vitro activity against *P. slundensis*, *E. coli*, and *S. aureus*, while 1,8-cineol only demonstrated activity against *E. coli* [68]. It can be concluded that cineol, widely recognized as an antibacterial agent [69,70,71,72,73], does not act via the same mechanism as the other compounds and therefore does not have the same enzymatic target. The binding energies of the phytocompounds were ranked in the following order: Native ligand > gentamicin > ellagic acid > epicatechin > α-terpineol > 1,8-cineol. Hydrogen bonds are essential for the interaction between inhibitors and receptors [28]. The absence of hydrogen bonds in cineol (No H bonds) resulted in its inactivity and demonstrated that its primary target differs from TyrRS.

### 4.4. Prediction of ADME/Toxicity and Similarity to Existing Drugs

An in silico analysis was performed to investigate the pharmacokinetic and toxicity properties of the compounds. Lipinski’s drug-likeness rules were met by all compounds without any violations, except gentamicin which had two violations. Furthermore, the compounds showed a high potential for absorption by the human intestine, which is in contrast to gentamicin, which has low gastrointestinal absorption and thus is administered in the injectable form [55]. The LogP and TPSA values indicate that the compounds have the potential for intramuscular, cutaneous, and intravenous administration. Notably, all compounds demonstrated good solubility, which is a critical factor for successful drug development, as poor solubility may affect xenobiotics’ pharmacokinetic and pharmacodynamic properties [74]. Regarding their safety profile on cytochromes P450 (1A2, 2C1, 2C9, 2D6, and 3A4), all phytoconstituents were deemed safe except for ellagic acid, which was found to be an inhibitor of CYP1A2. Despite this, Japan has approved it as an existing food additive [64]. P450 (CYP) enzymes are ubiquitous enzymes that play a crucial role in the metabolism of pharmaceuticals [75] and are involved in the activation and detoxification of endogenous and xenobiotic chemicals [76]. Interestingly, flavonoids have been shown to induce the biosynthesis of several CYPs, and the enzymatic activities of CYPs can be modulated (inhibited or stimulated) by these compounds [77].

## 5. Conclusions

Several factors, including availability, ease of preparation, safety, and affordability, drive herbal preparations’ consumption. In Algeria, *Eucalyptus* extracts are widely used in traditional medicine, but their use lacks standardization. The aqueous extract obtained from leaves of *E*. *radiata* and *E*. *cinerea* was tested for eventual bioactivities. In general, the results of this study indicated that the extracts possess potent antibacterial activity where *E*. *coli* and *S*. *aureus* were susceptible to the two Eucalyptus probably because of the presence of Ellagic acid and Epicatechin which showed strong TyRS bond energies. The extracts were found to possess radical scavenging on the DPPH radical and anti-inflammatory activities during albumin denaturation assay. In silico, results show a prominent binding of the ligand with the bacterial enzyme target. In addition, phytoconstituent docked were found to be free of cytotoxicity, supporting their future exploitation for oral or topical application. Thus, it can be concluded that aqueous extract of leaves of *E*. *radiate* and *E*. *cinerea* can be used as antibacterial, antioxidant, and anti-inflammatory agents or for the preparation of nutraceutical products and in food industries.

## Figures and Tables

**Figure 1 molecules-28-07153-f001:**
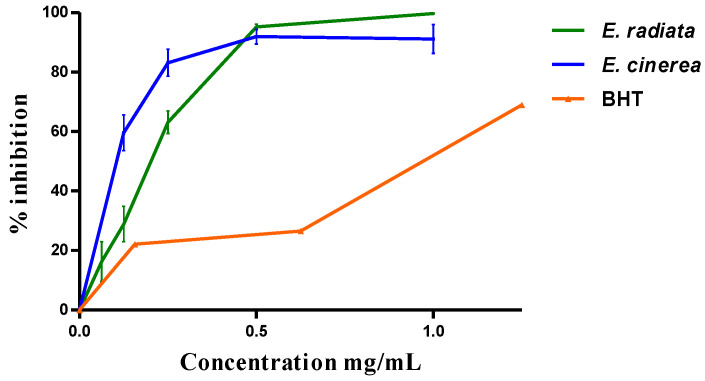
DPPH scavenging effect of the AEs and this of BHT.

**Figure 2 molecules-28-07153-f002:**
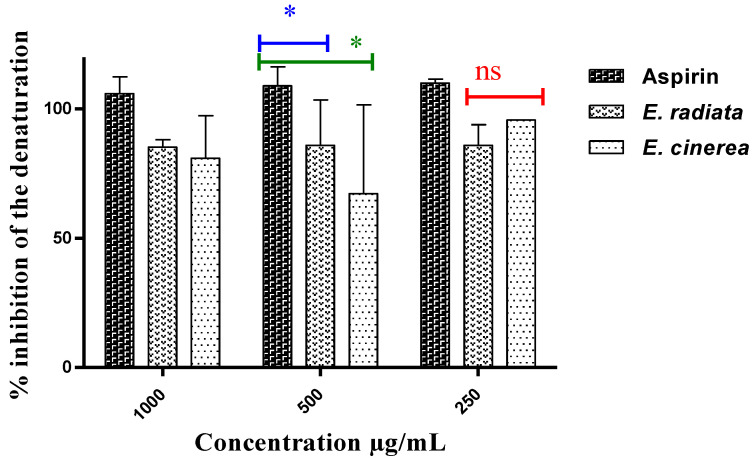
Inhibition of protein denaturation by the extracts and aspirin (*p* < 0.05), ns: non-significant, *: Slightly significant difference.

**Figure 3 molecules-28-07153-f003:**
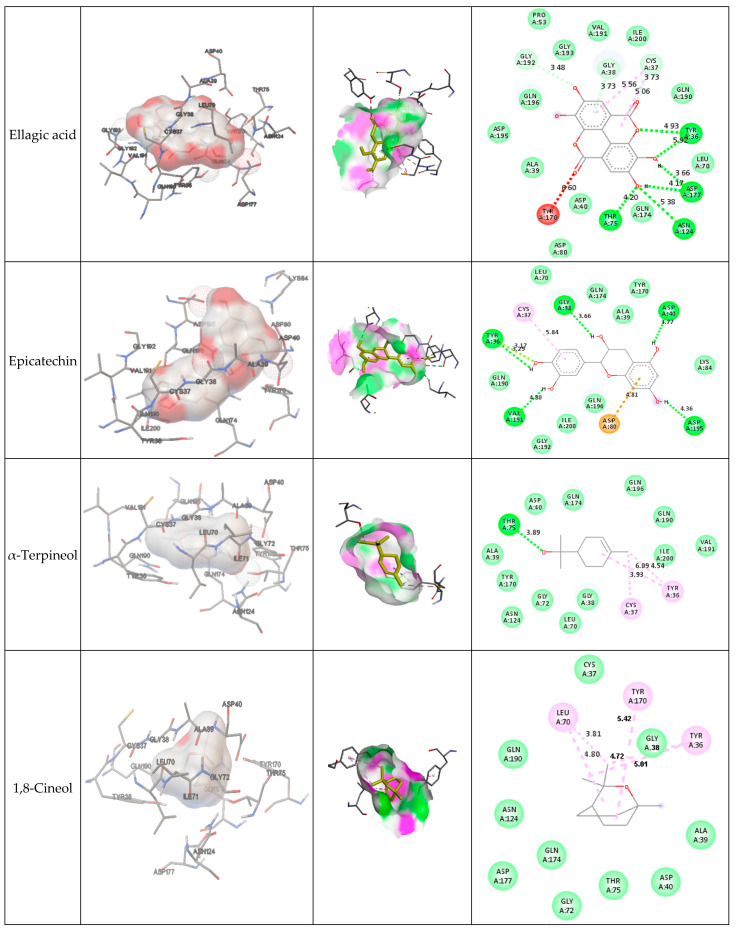
Interaction and bond distances of ligands inside the active site pocket as shown by molecular surface maps.

**Figure 4 molecules-28-07153-f004:**
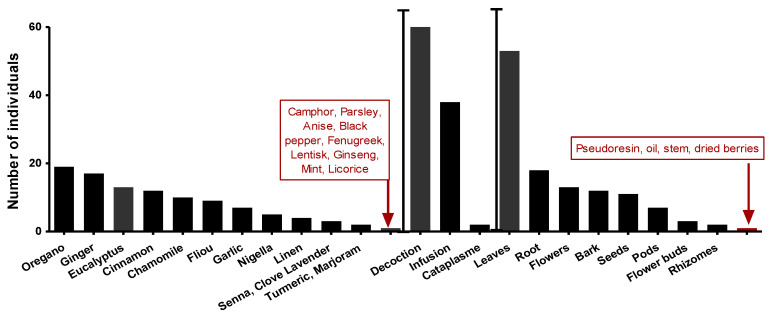
The most commonly used species, plant parts, and methods of preparation.

**Figure 5 molecules-28-07153-f005:**
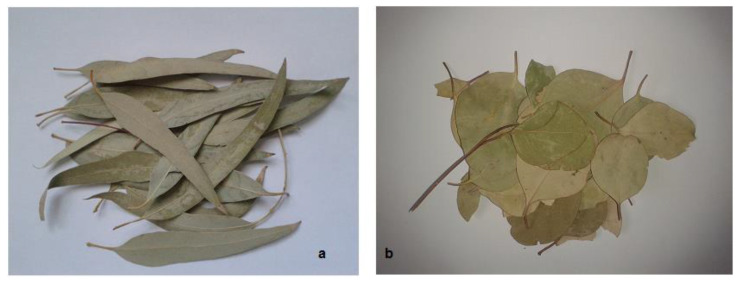
(**a**) *E*. *radiata*, (**b**) *E*. *cinerea*.

**Figure 6 molecules-28-07153-f006:**
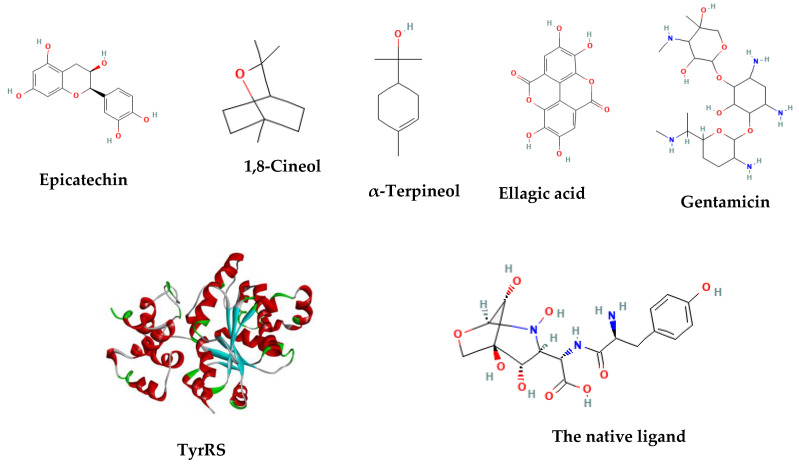
The TyrRS (1JII) crystal structure and the 2D structures of the tested compounds.

**Table 1 molecules-28-07153-t001:** Inhibition diameters (mm) of the extracts against the strains tested.

Plant	*E*. *radiata*				*E*. *cinerea*				GM
Concentration (mg/mL)	200	150	100	50	200	150	100	50	10 µg/disc
*E*. *coli*	20 ± 7.07	14 ± 1.00	13 ± 0	9 ± 00	19 ± 2.64	20 ± 0	20 ± 0	19 ± 1.73	40
*S*. *aureus*	18 ± 2	15.5 ± 0.70	16 ± 1.41	-	23.5	22.6 ± 2.51	18 ± 0	18.6 ± 1.15	40
*P*. *aeruginosa*	11.66 ± 1.52	12 ± 00	-	-	15.33 ± 0.57	12	-	-	27

-: no activity, GM: Gentamicin.

**Table 2 molecules-28-07153-t002:** Phytocompound scoring results.

Compound	Best Run	Free Energy of Binding (kcal/mol)	Inhibition Constant, Ki (μM)	vdW + H-Bond + Desolv Energy (kcal/mol)
Ellagic acid	8	−7.53	3.03	−8.48
Epicatechin	7	−6.75	11.31	−7.88
α-Terpineol	3	−6.43	19.31	−6.92
1,8-Cineol	7	−5.59	79.89	−5.55
Gentamicin	8	−10.04	44.00	−5.40
The native ligand	2	−10.42	23.10 nM	−10.88

**Table 3 molecules-28-07153-t003:** The amino acids involved in the active site of the TyrRS.

	H Bonds	VdW	C-H Bond	Pi-Alkyl Bonds	Alkyl Bonds	Pi-Anion
Ellagic acid-1TYA	Asp 243, Asp 244, Glu101, Glu 245	Ser45, Lys98, Lys246	Lys 105			
Epicatechin	Asp 40, Asp 195, Gly38, Tyr 36, Val 191	Leu70, Gln174, Ala39, Tyr170, Lys84, Gln196, Ile200, Gly192, Gln190		Cys 37		Asp 80
α-Terpineol-1TYA	Thr 75	Asp40, Gln174, Gln196, Gln190, Ile200, Val191, Gly38, Leu70, Gly72, Asn124, Tyr170, Ala39		Tyr 36, Cys 37	Tyr 36, Cys 37	
1,8-Cineol-1TYA		Cys37, Gly38, Ala39, Asp40, Thr75, Gly72, Gln174, Asp177, Asn124, Gln190.		Tyr 36, Tyr 170, Leu 70	Tyr 36, Tyr 170, Leu 70	
Gentamicin -1TYA	Glu 101, Asp 243, Asp 44, Glu 245	Lys246, Ser45, Lys98	Lys 105			
The native ligand	Glu 101, Asp 243, Asp 44, Glu 245	Ser 45, Lys 98, Lys 246	Lys 105			

**Table 4 molecules-28-07153-t004:** Calculated physicochemical and pharmacokinetic parameters of the docked phytocompounds.

Compound						
		Ellagic Acid	Epicatechin	α-Terpineol	1,8-Cineol	GM
Physicochemical and pharmacokinetic parameters (Molinspiration Cheminformatics)						
miLogP < 5		0.94	1.37	2.60	2.72	−4.21
TPSA (oA) < 500		141.33	110.37	20.23	9.23	199.74
MW < 500 (g/mol)		302.19	290.27	154.25	154.25	477.60
MV		221.78	244.14	170.65	166.66	450.66
nON < 10		8	6	1	1	12 (vio)
nOHNH < 5		4	5	1	0	11 (vio)
Lipinski’s violation		0	0	0	0	2
Solubility and pharmacokinetics properties (SwissADME)						
Water solubility		Soluble	Soluble	Soluble	Soluble	Highly soluble
Lipophilicity		Yes	Yes	Yes	Yes	
Gastrointestinal absorption		High	High	High	High	Low
Log Kp (skin permeation: cm/s)		−7.36	−7.82	−4.83	−5.30	−12.12
Cytochromes inhibitors	CYP1A2	Yes	No	No	No	No
	CYP2C19	No	No	No	No	No
	CYP2C9	No	No	No	No	No
	CYP2D6	No	No	No	No	No
	CYP3A4	No	No	No	No	No
Toxicity risks (OSIRIS Property Explorer)						
Mutagenic		No	No	No	Yes	No
Tumorigenic		No	No	No	No	No
Irritant		No	No	MR	No	No
Reproductive effective		No	No	No	Yes	No

miLogP: Logarithm of partition coefficient between n-octanol and water. TPSA: Topological polar surface area. MW: Molecular weight. MV: Molecular volume. nON: Number of hydrogen bond acceptors. nOHNH: Number of hydrogen bond donors. No: no indication found, MR: medium risk.

**Table 5 molecules-28-07153-t005:** The main phytocompounds found in aqueous extracts of *Eucalyptus* species.

Species	Compound	Extract	References
*E*. *radiata*	1,8-Cineol	The main compound in EOs of most species	[16,17]
*E*. *cinerea*	Ellagic acid Sideroxylonal B Macrocarpal A	Aqueous extract	[18]
		Aqueous extract	[3]
*E*. *camaldulensis*	Ellagic acid Gallic acid	Aqueous soluble fraction	[8]
*E*. *globulus*	1,8-Cineol α-Terpineol	Aqueous extract	[19]
	Ellagic acid Quercetin	Aqueous extract	[20]
	1,8-Cineol Epicatechin	Aqueous extract	[21]
	Ellagic acid	Hydrodistillation residual water	[22]
	Ellagic acid	Aqueous extract	[23]
*E*. *robusta*	Epicatechin Quercetin	Aqueous extract	[24]
*E*. *microcorys*	Ellagic acid Epicatechin	Aqueous extract	[25]
Different species	1,8-Cineol α-Terpineol	Aqueous Volatile Fractions	[26]

## Data Availability

Not applicable.

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
