# Peer review of "In Vitro and In Silico Activities of E. radiata and E. cinerea as an Enhancer of Antibacterial, Antioxidant, and Anti-Inflammatory Agents"

_molecules, 2023, doi:10.3390/molecules28207153_

Round 1
Reviewer 1 Report (New Reviewer)
The presented manuscript In vitro and in silico activities of E. radiata and E. cinerea as an enhancer of antibacterial, antioxidant, and anti-inflammatory agents is devoted to the urgent task - the quantitative evaluation of common herbal medicinal products, especially without animal testing.
The data obtained allow objective comparison of the medications used in medical practice among themselves and use these data to develop more effective complex medications.
The article presents exhaustive description of preparation of medicinal products from raw material - eucalyptus leaves, convincing quantitative assessment of the effectiveness of the obtained medicinal products as antibacterial, anti-inflammatory, and antioxidant medications as well as their toxicity.
It should be noted the use of a promising research method - the use of computer simulation of the binding process of chemical compounds in the decoction of eucalyptus leaves with the target molecule, in this case - TyrRS protein. This approach allows to determine the mechanism of biochemical action of the reagents under study and to relate the biochemical activity with their structure. Application of AI and IT methods allows to effectively investigate the biochemical activity of the medicinal products without using animal tests which, among other things, significantly accelerates and reduces the cost of the research process.
The data presented in the manuscript are new and original, the evidence for the claims given is adequate and exhaustive The manuscript is well structured, written clearly and logically.
However, the manuscript is not without its shortcomings. Both in the title and in the Abstract Section it is stated that eucalyptus leaves extract can be used to enhance the effectiveness of antibiotics. But there is no experimental evidence for this statement in the manuscript. I believe that either experimental evidence should be provided or the manuscript should be corrected.
Author Response
Attached

Reviewer 2 Report (New Reviewer)
Dear Authors,
the manuscript titled In vitro and in silico activities of E. radiata and E. cinerea as an enhancer of antibacterial, antioxidant, and anti-inflammatory agents discusses an important topic. In the investigation of antibacterial, antioxidant, and anti-inflammatory properties, two species belonging to the Myrtaceae family - E. radiata and E. cinerea were used. An aqueous extracts obtained from these plants showed the in vitro antibacterial activity. The most remarkable result was demonstrated by E. cenerea at 150 mg/mL against S. aureus. Both extracts displayed significant antioxidant activity compared to BHT. The anti-inflammatory impact was evaluated using a protein (BSA) inhibition denaturation test. The E. radiata extract was found to inhibit inflammation. Ellagic acid, a primary phytochemical found in the extracts, demonstrated noteworthy characteristics and a maximum binding energy of -7.53 kcal/mol for its anti-TyrRS activity in silico. The study suggests that the extracts and their primary phytochemicals could enhance the efficacy of antibiotics, antioxidants, and non-steroidal anti-inflammatory drugs (NSAIDs).
However, the manuscript requires very careful revision and correction. All comments and remarks are included in the attached file.

Round 2
Reviewer 2 Report (New Reviewer)
Dear Authors,
your improved manuscript In vitro and in silico activities of E. radiata and E. cinerea as an enhancer of antibacterial, antioxidant, and anti-inflammatory agents is now suitable for publication in Molecules.
This manuscript is a resubmission of an earlier submission. The following is a list of the peer review reports and author responses from that submission.
Round 1
Reviewer 1 Report
The manuscript described the use of extracts from Eucalyptus as an enhancer of antibacterial, antioxidant, and anti-inflammatory agents, and suggests that the extracts and their primary phytochemicals could be utilized to enhance the efficacy of existing antibiotics, antioxidants, and nonsteroidal anti-inflammatory drugs (NSAIDs). The reviewer believes that the manuscript cannot be accepted for publication in its current state for the following reasons:
It has been known for many years that eucalyptus leaves have many biological activities, and the main chemical components in eucalyptus leaves have been isolated and characterized in the literature. However, in this manuscript, only extracts were used for activity assay. Therefore, it is suggested that the author segment the eucalyptus leaf extracts to identify the effective sites that produce these activities, further isolate the active monomers, and conduct structural confirmation.
Author Response
Reviewer 1:
- It has been known for many years that eucalyptus leaves have many biological activities and the main chemical components in eucalyptus leaves have been isolated and characterized in the literature. However, in this manuscript, only extracts were used for activity assay. Therefore, it is suggested that the author segment the eucalyptus leaf extracts to identify the effective sites that produce these activities, further isolate the active monomers, and conduct structural confirmation.
Thank you for your valuable feedback on our manuscript. We appreciate your insightful comments and suggestions.
Regarding your concern about the use of Eucalyptus leaf extracts instead of identifying the active components, we would like to explain our rationale. As you mentioned, Eucalyptus leaves are known to have many biological activities, and their main chemical components have been extensively studied in the literature. However, the biological activities of different Eucalyptus species may vary, and the aqueous extract of E. radiata and E. cinerea, grown in Algeria, has not been tested before. We chose the aqueous extract on purpose because it is commonly used in decoction by the local population, which was confirmed by our previous survey.
Furthermore, our in-silico study was based on the molecules existing in the majority of Eucalyptus species, as reported in the literature. We agree that it is important to identify the effective sites that produce these activities and isolate the active monomers for structural confirmation. Therefore, we plan to conduct an HPLC analysis of these extracts as soon as possible to identify the active components.
We hope this explanation addresses your concern and shows our careful consideration in choosing the extract and our plan to further investigate the active components. We believe our study provides valuable insights into the biological activities of these eucalyptus species, which have not been extensively studied before.
Thank you again for your valuable comments.
Reviewer 2 Report
Comments
About molecular docking studies:
1. The pdb 3jii structure was not redocked with the crystallographic ligand
2. Was the protein protonated before the calculations?, if so, inform how it was performed, if not, the protonation must be performed and the calculations redone
3. What active site region was used?
4. Insert the result of the crystallographic ligand in table 3
5. Need to improve completion
6. In the summary line 34 the correct is "in silico"
Author Response
- About molecular docking studies:
- The pdb 3jii structure was not redocked with the crystallographic ligand
Thank you for your feedback on our manuscript. We have carefully reviewed your comments and made the necessary changes to our work. In particular, you mentioned that the pdb 1JII structure was not redocked with the crystallographic ligand, and we would like to clarify that we have now performed this analysis.
After considering your comment, we realized that this was an important step in our study, and we agree that it should have been included in our initial submission. Therefore, we conducted the redocking analysis using AutoDock Vina, and we obtained a binding mode similar to the crystallographic ligand. We have included the results of this analysis in the revised manuscript, along with an explanation of the methodology used.
We would like to thank you for your helpful comments, which have allowed us to improve the quality of our work. We hope that you find the revised manuscript satisfactory and look forward to your feedback.
- Was the protein protonated before the calculations? if so, inform how it was performed, if not, the protonation must be performed and the calculations redone
Thank you for your comments on our manuscript. Regarding your question about the protonation state of the protein in our calculations, we would like to inform you that the protein was indeed protonated before the calculations were performed.
To protonate the protein, we used the PDB2PQR web server, which is a widely used tool for protein preparation. This server generates the hydrogen atoms and assigns the protonation states of the ionizable residues based on the pH of the system. We set the pH of the system to 7.0, which is physiological pH, and the server assigned the protonation states of the residues accordingly.
We followed the standard protocol for protein preparation and performed energy minimization before conducting the docking simulations. The protein was then docked with the ligand using AutoDock Vina, and we analyzed the results to identify the binding modes and calculate the binding energies.
We hope that this information addresses your concern. If you have any further questions or comments, please do not hesitate to let us know. We appreciate your feedback and look forward to your response.
- What active site region was used?
Thank you for your comments on our manuscript. Regarding your question about the active site region used in our study, we would like to clarify that we performed a blind docking, which means that we did not define a specific active site region for our docking simulations.
In a blind docking, the docking software is allowed to explore the entire protein structure to identify potential binding sites for the ligand. This approach is useful when the binding site of the protein is not known or when there are multiple potential binding sites.
For our study, we used AutoDock Vina to perform the docking simulations, and the software was allowed to explore the entire protein structure to identify potential binding sites for the ligand. We then analyzed the results to identify the binding modes and calculate the binding energies.
We hope that this information clarifies the approach used in our study. If you have any further questions or comments, please do not hesitate to let us know. We appreciate your feedback and look forward to your response.
- Insert the result of the crystallographic ligand in table 3
Thank you for your suggestion regarding the inclusion of the crystallographic ligand result in Table 3 of our manuscript. We have now added the result to the table as requested.
The crystallographic ligand result is now included in the table along with the other ligand binding affinities calculated in our study. We have also included a note in the table to indicate which value corresponds to the crystallographic ligand result.
We appreciate your feedback and hope that the revised manuscript meets your expectations. If you have any further suggestions or comments, please do not hesitate to let us know.
- Need to improve completion
We have taken your feedback seriously, and we have made significant efforts to improve the quality of our work and the English language.
We have carefully revised the manuscript, paying close attention to the areas you highlighted in your previous comments. We have made changes to the text to address any issues with clarity and readability, and we have worked to ensure that our findings are presented accurately and comprehensively.
We would like to assure you that we are committed to producing high-quality research, and we take your comments and suggestions seriously. We hope that the revised manuscript meets your expectations and that you find our work to be of value.
Thank you for your time and attention to our research. We look forward to your response.
- In the summary line 34 the correct is "in silico"
We have reviewed the text and agree that the correct term is "in silico" rather than "in vitro" in the summary on line 34. We apologize for the mistake and have made the necessary corrections.
Thank you for bringing this to our attention, and we appreciate your careful review of our manuscript.
Reviewer 3 Report
The manuscript describes the usefulness of Algerian Eucalyptus, While the work described is of interest and utility, the following shortcomings must be addressed in a minor revision before the manuscript is acceptable for publication.
Comments to the Author:
(1) The paper needs to be edited by a native English speaker because the writing style is not reader-friendly. A few sentences are not clearly addressed and are ambiguous.
(2) Page 2; line 85, I would like the authors to confirm whether is it "Expensive" or "inexpensive"
(3) In Figure 3, the 2d structures are a little blurry, improving the resolution of the structures is recommended.
Author Response
- The paper needs to be edited by a native English speaker because the writing style is not reader-friendly. A few sentences are not clearly addressed and are ambiguous.
Thank you for your feedback on our manuscript. We appreciate your comments regarding the writing style and clarity of our manuscript. We agree that the text could benefit from further editing by a native English speaker to improve the overall readability and clarity.
We have taken your feedback into consideration and have worked to revise the manuscript to ensure that it is more reader-friendly and that any ambiguous sentences are clarified. We recognize the importance of clear and concise language in scientific writing, and we have made efforts to meet these standards in our revised manuscript.
We would like to thank you for your time and attention to our work, and we hope that the revised manuscript meets your expectations.
- Page 2; line 85, I would like the authors to confirm whether is it "Expensive" or "inexpensive"
Thank you for your feedback on our manuscript. We appreciate your comment regarding the cost of the method mentioned on page 2, line 85. To clarify, the method we used is relatively inexpensive compared to other experimental methods used to identify potential drug candidates.
We apologize for any confusion caused by the original wording and have made the necessary revisions to clarify this point in the revised manuscript.
Thank you for bringing this to our attention, and we appreciate your careful review of our work.
- In Figure 3, the 2d structures are a little blurry, improving the resolution of the structures is recommended.
Dear Reviewer, thank you for your comment on our manuscript. We have carefully reviewed Figure 3 and agree that the 2D structures were blurry. We have made efforts to improve the resolution of the structures to make them clearer and more readable.
We appreciate your attention to detail and your helpful feedback in improving the quality of our manuscript. Thank you for your time and effort in reviewing our work.
Round 2
Reviewer 1 Report
Before acceptable for publication, it is necessary to conduct an HPLC analysis of these extracts to identify the active components.
Author Response
Reviewer 1:
- Before acceptable for publication, it is necessary to conduct an HPLC analysis of these extracts to identify the active components.
Regarding the suggestion to conduct HPLC analysis, we regret to inform you that it is not possible for us to perform this analysis due to legal restrictions and resource limitations. In our laboratory, several research teams work together, and submitting the same work for publication is prohibited by law. Furthermore, another research team has already conducted the HPLC analysis and submitted it for publication. Therefore, we have decided to divide our work into two parts. The first part involves testing the in vitro activities of the extracts and predicting their antibacterial activity and toxicity. The second part involves extracting the pure compounds from the extracts separately and testing them individually in vivo to limit the number of animals used. We believe that these two parts of our study will provide valuable insights into the activities of E. radiata and E. cinerea.
Reviewer 2 Report
Protein protonation information must be entered in the docking section
Author Response
Reviewer 2
- Protein protonation information must be entered in the docking section
Thank you for your feedback regarding the protein protonation information in the docking section. We have carefully considered your suggestion and have made the necessary updates to include this information in our manuscript. We appreciate your valuable input and attention to detail, and we believe that this update will enhance the scientific rigor and clarity of our study. Thank you again for your time and efforts in reviewing our manuscript.